# Acetyl-CoA Carboxylase Inhibitor CP640.186 Increases Tubulin Acetylation and Impairs Thrombin-Induced Platelet Aggregation

**DOI:** 10.3390/ijms222313129

**Published:** 2021-12-04

**Authors:** Marie Octave, Laurence Pirotton, Audrey Ginion, Valentine Robaux, Sophie Lepropre, Jérôme Ambroise, Caroline Bouzin, Bruno Guigas, Martin Giera, Marc Foretz, Luc Bertrand, Christophe Beauloye, Sandrine Horman

**Affiliations:** 1Pôle de Recherche Cardiovasculaire (CARD), Institut de Recherche Expérimentale et Clinique (IREC), Université Catholique de Louvain (UCLouvain), 1200 Brussels, Belgium; marie.octave@uclouvain.be (M.O.); laurence.pirotton@uclouvain.be (L.P.); audrey.ginion@uclouvain.be (A.G.); valentine.robaux@uclouvain.be (V.R.); sophie.lepropre@gmail.com (S.L.); luc.bertrand@uclouvain.be (L.B.); christophe.beauloye@uclouvain.be (C.B.); 2Centre de Technologies Moléculaires Appliquées, Institut de Recherche Expérimentale et Clinique (IREC), Université Catholique de Louvain (UCLouvain), 1200 Brussels, Belgium; jerome.ambroise@uclouvain.be; 3IREC Imaging Platform, Institut de Recherche Expérimentale et Clinique (IREC), Université Catholique de Louvain (UCLouvain), 1200 Brussels, Belgium; caroline.bouzin@uclouvain.be; 4Department of Parasitology, Leiden University Medical Center, 2333 ZA Leiden, The Netherlands; B.G.A.Guigas@lumc.nl; 5Department of Molecular Cell Biology, Leiden University Medical Center, 2333 ZA Leiden, The Netherlands; m.a.giera@lumc.nl; 6CNRS, INSERM, Institut Cochin, Université de Paris, F-75014 Paris, France; marc.foretz@inserm.fr; 7Division of Cardiology, Cliniques Universitaires Saint-Luc, 1200 Brussels, Belgium

**Keywords:** acetyl-CoA carboxylase, platelet, platelet aggregation, acetylation, tubulin, actin cytoskeleton, Rac1 signaling

## Abstract

Acetyl-CoA carboxylase (ACC) is the first enzyme regulating de novo lipid synthesis via the carboxylation of acetyl-CoA into malonyl-CoA. The inhibition of its activity decreases lipogenesis and, in parallel, increases the acetyl-CoA content, which serves as a substrate for protein acetylation. Several findings support a role for acetylation signaling in coordinating signaling systems that drive platelet cytoskeletal changes and aggregation. Therefore, we investigated the impact of ACC inhibition on tubulin acetylation and platelet functions. Human platelets were incubated 2 h with CP640.186, a pharmacological ACC inhibitor, prior to thrombin stimulation. We have herein demonstrated that CP640.186 treatment does not affect overall platelet lipid content, yet it is associated with increased tubulin acetylation levels, both at the basal state and after thrombin stimulation. This resulted in impaired platelet aggregation. Similar results were obtained using human platelets that were pretreated with tubacin, an inhibitor of tubulin deacetylase HDAC6. In addition, both ACC and HDAC6 inhibitions block key platelet cytoskeleton signaling events, including Rac1 GTPase activation and the phosphorylation of its downstream effector, p21-activated kinase 2 (PAK2). However, neither CP640.186 nor tubacin affects thrombin-induced actin cytoskeleton remodeling, while ACC inhibition results in decreased thrombin-induced reactive oxygen species (ROS) production and extracellular signal-regulated kinase (ERK) phosphorylation. We conclude that when using washed human platelets, ACC inhibition limits tubulin deacetylation upon thrombin stimulation, which in turn impairs platelet aggregation. The mechanism involves a downregulation of the Rac1/PAK2 pathway, being independent of actin cytoskeleton.

## 1. Introduction

Acetyl-CoA carboxylase (ACC) catalyzes the first step of de novo lipogenesis (DNL) by carboxylating acetyl-CoA into malonyl-CoA [1]. There are two ACC isoforms, ACC1 and ACC2, which are differentially distributed amongst cells and tissues [2]. ACC1 provides a pool of cytosolic malonyl-CoA, which serves as a substrate for fatty acid synthase and chain elongation systems. ACC2 produces malonyl-CoA near the mitochondrial membrane where it acts as an allosteric inhibitor of carnitine palmitoyltransferase 1 (CPT1); this enzyme is responsible for transporting long-chain fatty acids into mitochondria [3,4,5]. ACC activity is acutely controlled by both allosteric modulators and the phosphorylation of several serine residues [5,6]. We previously demonstrated that ACC1 is the predominant isoform in murine and human platelets [7]. ACC1 promotes thrombosis through platelet phospholipid production, which is required for the synthesis of arachidonic acid (AA), a key mediator of platelet activation [7]. More recently, the presence of ACC in DNL has also been implicated in megakaryocyte (MK) maturation and platelet production [8]. ACC’s critical role in megakaryocytes (MKs) and platelets raises the question of a potential therapeutic usefulness of prescribing pharmacological ACC inhibitors for a plethora of diseases characterized by DNL dysregulation [9,10,11,12], including Type 2 diabetes, cancer, or non-alcoholic fatty liver disease [10,13].

ACC inhibition not only decreases lipogenesis, but concomitantly results in elevated cytoplasmic levels of acetyl-CoA, which serves as a substrate for protein acetylation [14,15]. The latter process is a posttranslational modification that is regulated by lysine acetyltransferases (KATs) and histone deacetylases and sirtuins (HDACs, SIRTs) [16]. In platelets, protein acetylation on lysine residues is catalyzed by KATs P300 and α-TAT [17,18]. Deacetylation is induced by lysine deacetylases HDAC6 and SIRT2 [19,20]. Among acetylable proteins, as identified in platelets using proteomic studies [17], tubulin is recognized to play a crucial role in regulating platelet formation and functions [19,21,22,23]. MKs and platelets express at least six isoforms of α and β-tubulin. In resting platelets, stable microtubules that form the marginal band are composed of α-and β-tubulin heterodimers, which are highly acetylated (on Lys40 residue of α-tubulin) so as to preserve the discoid shape [24,25,26]. Upon agonist stimulation, α-tubulin is massively deacetylated by HDAC6, which results in the condensation of microtubules towards the platelets’ center, where they undergo de-polymerization [19]. This process is followed by their fragmentation and re-polymerization in order to generate a new microtubule coil [24,25,26]. According to several studies, this deacetylation process coordinates cytoskeletal and signaling events upon platelet activation, which in turn promote platelet spreading and aggregation [19,27].

In this study, we focused on the impact of the pharmacological ACC inhibitor, CP640.186, on the acetylation status of tubulin in human platelets. In addition, we examined the effect of ACC inhibition on platelet aggregation and secretion, as well as on cytoskeleton signaling and organization. We have herein demonstrated for the first time that ACC inhibition limits tubulin deacetylation upon thrombin stimulation, thereby impairing platelet aggregation. The molecular mechanism involves a downregulation of the Rac1/PAK2 pathway, while being independent of actin cytoskeleton. Similar results were obtained with tubacin, an inhibitor of HDAC6. Moreover, we have detected a decrease in thrombin-induced reactive oxygen species (ROS) production and extracellular signal-regulated kinase (ERK) phosphorylation in CP640.186-treated platelets. These findings suggest that ACC inhibitors may possibly alter platelet functions through a mechanism that is similar, at least to some extent, to that of HDAC inhibitors, which are well recognized to result in thrombocytopenia and hemostasis defects [28,29].

## 2. Results

### 2.1. Short-term ACC Inhibition Using CP640.186 Does Not Reduce Lipid Content in Human Platelets and Preserves Mitochondrial Function

We first investigated whether short-term pharmacological ACC inhibition would be able to reduce DNL in human platelets. Platelets were incubated with [1-^14^C]-acetic acid and treated for 2 h with DMSO or CP640.186 (60 µM), the latter being an isozyme-nonselective inhibitor of mammalian ACC. As a result, fatty acid synthesis was drastically reduced in CP640.186-treated platelets (Figure 1a). To evaluate the consequences of CP640.186 treatment on overall platelet lipid content, we undertook a quantitative lipidomic analysis of dimethyl sulfoxide (DMSO)- and CP640.186-treated platelets. Overall, 672 lipids were detected, including different phospholipids (phosphatidylethanolamine [PE], phosphatidylethanolamine plasmalogen [PEP], plasmanyl phosphatidylethanolamine [PEO], phosphatidylcholine [PC], lysophosphatidylethanolamine [LPE], lysophosphatidylcholine [LPC]), sphingomyelin [SM], cholesteryl esters [CE], free fatty acids [FFA], diacylglycerol [DAG], and triacylglycerol [TAG]) (Figure 1b). Paradoxically, a slight increase in 63 lipids belonging to TAG, LPC, and LPE lipid classes was observed following CP640.186 treatment (Figure 1b; Appendix A). However, lipid variations were low, consisting of a maximum 1.25-fold change, which indicates that a 2-h CP640.186 treatment does not drastically modify overall platelet lipid composition.

Next, we sought to investigate whether pharmacological ACC inhibition within platelets actually affected platelet bioenergetics, which was assessed by measuring mitochondrial function. As illustrated in Figure 1c,d, extracellular flux analysis revealed that the platelets’ basal oxygen consumption rate (OCR) was similar between DMSO- and CP640.186-treated platelets, and it was comparable to data published in the literature, as well [30]. The oxidative phosphorylation component was further assessed by injecting oligomycin, which is a complex V inhibitor. This led to an expected decrease in OCR. ATP-linked respiration was similar under both conditions. An injection of carbonyl cyanide p-trifluoromethoxyphenylhydrazone, a proton-ionophore, induced maximal OCR. DMSO- and CP640.186-treated platelets displayed a similar maximal OCR and reserve capacity. Finally, antimycin A and rotenone, which are complex III/I inhibitors, totally inhibited mitochondrial-induced OCR. The non-mitochondrial OCR and proton leak remained unchanged (Figure 1c,d). These data confirm that, in platelets, ACC1 does not play a major role in mitochondrial respiration.

### 2.2. CP640.186 Increases α-Tubulin Acetylation Level and Impairs Platelet Aggregation

To examine the impact of CP640.186 on platelet α-tubulin Lys 40 acetylation, we performed western blot analysis of α-tubulin deacetylation upon thrombin stimulation. As seen in Figure 2a, platelet pretreatment using CP640.186 increased α-tubulin acetylation levels at basal state and after thrombin stimulation. Next, we investigated the consequences of ACC inhibition on platelet functions. CP640.186 inhibited platelet aggregation in response to low thrombin doses (Figure 2b). A similar effect was observed using tubacin, an HDAC6 inhibitor causing a drastic increase in α-tubulin acetylation (Appendix A and Figure 2c). Impaired platelet aggregation was unlikely accounted for altered platelet secretion processes, given that neither CP640.186 nor tubacin decreased P-selectin surface exposure or ATP secretion upon thrombin stimulation (Figure 2d,e and Appendix A). Moreover, ACC inhibition did not affect thrombin-induced platelet αIIbβ3 activation, as measured by the binding of PAC-1 to platelets (Figure 2f). A comparable result was obtained with tubacin (Appendix A).

### 2.3. CP640.186 Does Not Affect Platelet Microtubule Dynamics

Given the impact of α-tubulin acetylation on microtubule dynamics in resting and collagen-related peptide (CRP)-activated platelets [19], we postulated that an altered microtubule structure could explain the effect of CP640.186 on thrombin-induced platelet aggregation. To assess the impact of CP640.186-induced tubulin acetylation on microtubules, washed human platelets were treated with or without the ACC inhibitor; then, they were fixed, plated onto fibrinogen-coated coverslips, and incubated with specific antibodies targeting both α-tubulin and acetylated α-tubulin (K40) (Figure 3a). In DMSO-treated platelets, the marginal band was observed as a continuous ring structure, which was not altered by CP640.186 pre-incubation (Figure 3a; white arrow). Next, we examined tubulin condensation in platelets plated onto fibrinogen-coated coverslips for 10 min, with or without thrombin, then fixed and stained as described above. In DMSO- and CP640.186-treated platelets, the sole activation by fibrinogen induced the coiling of the microtubule ring, in parallel of α-tubulin deacetylation (Figure 3b). Thrombin-induced platelet activation further promoted the microtubules’ remodeling by inducing their de-polymerization and formation of a new flat and de-acetylated ring at platelet periphery, in addition to the small one observed in the platelet center (Figure 3c, white arrow). The degree of platelet α-tubulin condensation did not differ between DMSO- and CP640.186-treated platelets (Figure 3d,f), indicating that ACC inhibition and the subsequent increase in α-tubulin acetylation did not result in obvious tubulin cytoskeleton changes in resting and thrombin-stimulated platelets. These conclusions were supported by results showing that tubacin, which drastically increases tubulin acetylation, did not alter the dynamics of tubulin organization either in resting or thrombin-activated platelets (Appendix A).

### 2.4. CP640.186 Reduces the Activation of Rac1-PAK2 Pathway in Response to Thrombin, without Affecting Actin Cytoskeleton Remodeling

Next, we hypothesized that the increased acetylation state of platelet α-tubulin could regulate specific signaling pathways activated downstream of protease-activated receptor 1 (PAR1) or PAR4, and that this mechanism could contribute to the partial disruption of platelet aggregation induced by ACC or HDAC6 inhibition. Aslan et al. previously reported that HDAC6 influences signaling upstream of the Rho GTPase Rac and its effector, p21-activated kinase (PAK) [19]. Both proteins likely appear to be critical players in platelet activation downstream of PAR receptors [31]. Importantly, Rac1 and PAK2 have previously been implicated in regulating actin polymerization in platelets, MKs, [31,32] and other cell types [33,34,35]. We sought to better understand whether this pathway’s downregulation could contribute to the reduced aggregation as observed in CP640.186-treated platelets. To address this issue, we investigated the CP640.186 effect on Rac1 activation, PAK2 phosphorylation, and actin polymerization in thrombin-stimulated platelets. Thrombin caused a rapid and sustained activation of Rac1, with maximal activity reached within 30 s (Figure 4a). Thrombin-induced Rac1 activity was reduced in CP640.186-treated platelets (Figure 4a), as was PAK2 phosphorylation (Figure 4b). Comparable results were obtained with tubacin (Figure 4c,d). Since the Rac1-PAK pathway tightly regulates actin cytoskeletal polymerization, we investigated the CP640.186 effect on the phosphorylation state of cofilin, which represents a downstream cytoskeletal target that is involved in actin turnover regulation. In platelets, a rapid first phase of cofilin de-phosphorylation/activation is detected, which is then followed by a second phase consisting of LIM domain kinase 1 (LIMK1)-mediated slow cofilin re-phosphorylation/inactivation [36,37]. As seen in Figure 4e, CP640.186 pre-incubation did not influence the cofilin de-phosphorylation/phosphorylation cycle in response to thrombin. Accordingly, thrombin-induced F-actin formation was not altered in CP640.186-treated platelets (Figure 4f). To confirm that actin cytoskeleton was not impacted by ACC inhibition, nor by the subsequent alteration of thrombin-induced Rac1-PAK2 signaling, we additionally examined filopodia, lamellipodia, and stress fiber formation within platelets plated onto fibrinogen-coated coverslips and stimulated with thrombin for 15 min. Under thrombin stimulation, most platelets were fully spread and actually formed stress fibers. CP640.186 treatment did not change the proportion of platelets containing filopodia, lamellipodia, or stress fibers (Appendix A). Again, the impact of tubacin on cofilin phosphorylation and actin polymerization/organization was comparable to that of CP640.186 (Appendix A). This observation clearly supports that, under these experimental conditions, the inhibitory effect of CP640.186 on platelet aggregation is most likely due to the increased α-tubulin acetylation level without being caused by altered actin cytoskeleton remodeling.

### 2.5. CP640.186 Alters Thrombin-Induced ROS Production and ERK Phosphorylation

Rac1 and PAK likely affect downstream signaling events that are unrelated to actin cytoskeleton. For instance, Rac1 is an integral part of ROS generation by NADPH oxidase isoforms NOX1 and NOX2 [38,39]. To test whether ACC inhibition did alter ROS production, platelets were incubated with 2’,7’-dichlorodihydrofluorescein diacetate (H_2_DCFDA) probe. Consistent with Rac1′s role in NOX2 activation, CP640.186-treated platelets exhibited a significant decrease in basal and thrombin-induced ROS production (Figure 5a). Similar results were obtained in tubacin-treated platelets (Appendix A). PAK also displays non-cytoskeletal substrates, as demonstrated by the mitogen-activated protein kinase (MAPK) activation through phosphorylation and activation of MEK proteins [31,40]. Accordingly, the downstream MEK-driven phosphorylation of ERK was reduced in CP640.186-treated platelets following thrombin stimulation (Figure 5b). However, under these experimental conditions, we were unable to reveal any tubacin impact on ERK phosphorylation (Appendix A).

## 3. Discussion

ACC is a key enzyme of lipid synthesis and oxidation pathways, rendering it an attractive target for various metabolic diseases [9,10,11,12]. Over the last years, several ACC inhibitors have entered various stages of clinical testing or have already been approved for clinical use [41]. Moreover, this new class of therapeutics has been stated to share unexpected clinical implications, including the potential to lower platelet counts in humans and nonhuman primates [8]. Regarding platelet functions, the potential impact of these molecules remains elusive. In this study, we examined in vitro the impact of ACC inhibition on platelet activation, while investigating the underlying mechanisms. Using washed human platelets, we have demonstrated for the first time that a short-term incubation with CP640.186, which is a nonselective, reversible, and ATP-noncompetitive inhibitor of ACC1/2 [42], increases α-tubulin acetylation levels in both resting states and under thrombin-stimulated conditions. This was associated with impaired platelet aggregation. The molecular mechanism connects α-tubulin acetylation to the down-regulation of Rac1-PAK2 signaling and decreased ROS production.

The first publications reporting a beneficial impact of ACC inhibition on various metabolic diseases originated from studies investigating the long-chain fatty acid analog, 5-(tetradecyloxy)-2-furoic acid (TOFA) [43]. However, the similarity of TOFA to various long-chain fatty acids raised the issue as to the specificity of this drug with respect to ACC, and this in relation to the inhibition of other long-chain fatty acyl-CoA-utilizing enzymes [42,44]. CP640.186 binds ACC within the carboxyltransfer domain, and the agent has been identified by Pfizer researchers through high-throughput screening [42]. Its binding site appears to be distinct from that found on other enzymes that catalyze carboxyltransfer reactions, including pyruvate carboxylase and propionyl-CoA carboxylase, which renders this drug highly specific [45]. This compound was shown to inhibit DNL in hepatic cells [42]. In human platelets, CP640.186 treatment for a maximum of 2 h was associated with DNL decreases without drastically affecting overall platelet lipid composition. Several lipid species among TAG, LPC, and LPE classes were even found to be increased. Similar paradoxical findings have been reported in neuroblastoma cells in which TOFA-induced DNL inhibition resulted in an elevation of long chain fatty acids (tetracosanoic acid [C24:0]) and of some sphingomyelins, as well [46]. Regarding fatty acid utilization to produce ATP, our results clearly showed that mitochondrial respiration is not affected by pharmacological ACC inhibition.

In an earlier study, a lower ACC1 expression in yeast was demonstrated to be associated with the hyperacetylation of several proteins, including histones and cytoplasmic proteins that are involved in metabolism [15]. More recently, the impact of ACC inhibition on protein acetylation was found to be linked to cancer cells’ metastatic capacity [14]. The present study is the first to connect ACC inhibition and protein acetylation within human platelets. The results clearly showed CP640.186, similarly to HDAC6-specific inhibitor tubacin, to be able to limit thrombin-induced deacetylation of tubulin, thereby inhibiting platelet aggregation. The acetylation of α-tubulin has been recognized to play a key role in microtubule organization and stability [19,47,48]. However, following the immobilization of CP640.186-treated platelets on fibrinogen-coated coverslips, we were unable to identify any treatment impact on the marginal band of resting platelets, nor on their ability to condense upon fibrinogen and thrombin stimulation. Platelet spreading and morphology were also preserved. These results are supported by similar observations reported in tubacin-treated human platelets or in platelets isolated from HDAC6 knockout mice [18,27]. In contrast, Aslan et al. did not reach the same conclusions. Indeed, based on the visualization of tubacin-treated human platelets using super-resolution structured illumination microscopy, these authors described a disrupted marginal band at resting state, along with a reduced ability to condense upon CRP stimulation [19].

Additionally, CP640.186 and tubacin were shown to reduce PAK2 phosphorylation downstream of Rac1 GTPase activation, although the molecular link between microtubule acetylation and Rac1 is still poorly understood. In neurons, it has been proposed that p120 catenin associates with and stabilizes microtubules, which promotes Rac1 activation and neurite outgrowth [49]. Interestingly, p120-catenin is present in platelets and likely influences Rho-GTPases activity and key signaling molecules involved in platelet aggregation, including E-cadherin [50]. The potential contribution of p120-catenin to the mechanism by which CP640.186 regulates Rac1 in platelets warrants further investigations. Once activated, Rac1 is able to modulate different enzymatic activities. For instance, Rac1 plays a crucial role in activating Nox family NADPH oxidase enzymes dedicated to ROS production, including superoxide anion radical or hydrogen peroxide. NOX2 is activated in a Rac1-dependent manner. In the GTP-bound form, Rac1directly binds to the oxidase activator p67 (PHOX), which in turn interacts with NOX2, leading to superoxide anion production [39]. Consistent with the CP640.186 effect on Rac1 activity, ROS generation was significantly decreased in response to thrombin stimulation.

PAK mediates the phosphorylation of a plethora of proteins involved in actin cytoskeleton, such as LIMK1, merlin, or myosin light chain kinase (MLCK) [34]. In platelets, the Rac1-PAK signaling cascade has been shown to regulate actin polymerization via LIMK1-cofilin phosphorylation and lamellipodia formation [31,40,51]. However, we did not observe any differences in thrombin-induced cofilin (de)phosphorylation, shape alterations, or F-actin content between DMSO- and CP640.186-treated platelets. Accordingly, platelets from HDAC6 KO mice displayed normal platelet shape changes, which let us assume that acetylated α-tubulin is not implicated in the regulation of actin cytoskeleton [18,27]. Interestingly, PAK is able to activate substrates other than proteins regulating cytoskeleton organization. For instance, ERK is rapidly phosphorylated in activated platelets. This phosphorylation notably depends on PAK [31,34,40], and it has been shown to play a role in platelet aggregation and thrombus formation in a laser-injured subcutaneous arteriole model [52,53]. Mechanistically, ERK activation has been reported to constitute a key signaling event designed to expose on platelet surface adhesion molecules that are necessary for platelet tethering. The latter constitutes the first step of platelet aggregation [52]. The impaired thrombin-induced aggregation of CP640.186-treated platelets possibly results from a similar mechanism. Moreover, our data are in line with studies showing that tubacin-induced increases in α-tubulin acetylation levels were associated with a decrease in CRP-induced ERK phosphorylation [19,54]. We failed to detect any significant decrease in ERK phosphorylation in thrombin-stimulated, tubacin-incubated platelets. This observation suggests that ACC and HDAC6 inhibitions likely exhibit different downstream effects. A limitation of our study is that we only correlate α-tubulin acetylation with the inhibition of platelet aggregation and Rac1/PAK2 signaling. We cannot exclude that other acetylated proteins may contribute to the effects of both CP640.186 and tubacin.

In summary, our data demonstrate that pharmacological ACC inhibition is able to impair thrombin-induced platelet aggregation, an effect which is associated with an increase in α-tubulin acetylation levels. The underlying mechanisms involve a downregulation of the Rac1-PAK2 signaling pathway, which is associated with a decrease in ROS production and ERK phosphorylation. Our work adds momentum to the emerging notion that precautions would be needed for studies using ACC inhibitors in Type 2 diabetes, cancer, or nonalcoholic steatohepatitis. Reducing platelet activation may likewise exert positive effects, notably in some inflammatory or thrombotic disorders. Future research investigating in vivo the impact of DNL inhibitors on platelet functions are now needed, given that new molecules are currently entering clinical development.

## 4. Materials and Methods

### 4.1. Reagents

CP640.186 (#17691) was purchased from Cayman Chemical (Ann Arbor, Michigan, United States of America [USA]). Thrombin from bovine plasma (#T6634), apyrase (#A6132), fibrinogen (#F4129), fluorescein isothiocyanate (FITC)—conjugated phalloidin (#P5282), sigmacote (#SL2), Dulbecco′s Modified Eagle′s Medium (DMEM, #D5030), prostaglandin E1 (PGE1) (#P5515), and horse radish peroxidase (HRP)- conjugated anti-rabbit (#A0545) antibodies (Abs) were purchased from Sigma-Aldrich (Overijse, Belgium). [1-^14^C]-Acetic Acid was obtained from Perkin Elmer (#NEC084H001MC) (Zaventem, Belgium). S-MONOVETTE CPDA tube (#11610001) was from Sarstedt (Etten-Leur, The Netherlands). Eptifibatide was obtained from GlaxoSmithKline (Wavre, Belgium). Bovine serum albumin (BSA) (#3854.4) was purchased from Roth (Keerbergen, Belgium). Fatty acid-free BSA (#126575) was from Millipore (Overijse, Belgium). Anti-tubulin antibody (#62204) and H_2_DCFDA probe (#D-399) were purchased from Thermo-Fisher Scientific (Waltham, MA, USA). Complete, Mini Protease Inhibitor Cocktail (#11836153001) was obtained from Roche (Brussels, Belgium). Luciferin-luciferase reagent (#CH395) was from Stago BNL (Mont-Saint-Guibert, Belgium), and Cell Tak (#354240) was from Corning (Lasne, Belgium). Oligomycin, FCCP, antimycin-A, and rotenone were from Agilent (Mito Stress Test Kit, #103015-100) (Santa Clara, CA, USA). Rac1 G-LISA kit (#BK128) was from Cytoskeleton, Inc, (Denver, CO, USA). Anti-phospho-cofilin (#3313), anti-cofilin (#5175), anti-phospho-ERK (Thr202/Tyr204) (#9101), anti-ERK (#9102), anti-phospho-PAK (Ser199/204; Ser 192/197) (#2605), anti-PAK2 (#2615), anti-acetyl-α-tubulin (#5335), anti-GAPDH (#2118), anti-rabbit IgG HRP-linked (#7074), and anti-gelsolin (#12953) antibodies were obtained from Cell Signaling (Danvers, MA, USA). Phycoerythrin-conjugated anti-CD62P (#555524), FITC-conjugated anti-active αIIbβ3 (PAC-1) (#340507), HRP-conjugated anti-mouse (#554002), donkey anti-mouse IgG coupled with alexa fluor 594 (#A-21203), and donkey anti-rabbit IgG coupled with alexa fluor 488 (#A-21206) antibodies were purchased from BD Bioscience (San Jose, CA, USA).

### 4.2. Human Platelets Preparation

Venous blood was taken from human healthy volunteers (without drug intake one week before blood sampling) having provided their consent in citrated tubes (1/10; citrate, phosphate, dextrose, adenine). Platelet rich plasma (PRP) was obtained by centrifugation at 330× *g* for 20 min. After adding eptifibatide (4 µg/mL) and apyrase (1 U/mL), platelets were pelleted by a centrifugation at 800× *g* for 10 min. Platelets were washed once with a modified Tyrode’s buffer (135 mM NaCl, 12 mM NaHCO_3_, 3 mM KCl, 0.3 mM Na_2_HPO_4_, 1 mM MgCl_2_, 5 mM D-glucose, 10 mM Hepes, and 0.35% BSA, pH 7.4, 37 °C) containing eptifibatide (4 µg/mL) and apyrase (1 U/mL) at 1000× *g* for 10 min. Platelets were re-suspended in modified Tyrode’s buffer to a density of 2.5 × 10^5^/µL (unless said otherwise). Washed platelets were incubated 2 h at 37 °C with CP640.186 before platelet function assays.

### 4.3. Lipogenesis Measurement

Platelets were re-suspended in a modified Gaintner buffer (4.92 mM KH_2_PO_4_, 40 mM NaH_2_PO_4_.H_2_O, 102.7 mM NaCl, 40 mM NaOH, and 5.5 mM D-glucose, pH 6.5, 37 °C) at a density of 6.5 × 10^5^/µL. Washed platelets (6.5 × 10^8^) were incubated for 2 h at 37 °C in a solution including 1mL of modified Gaintner buffer, 4.15 mM acetate, and 4.415 µCi/µmol [1-^14^C]-acid acetic. To stop the reaction, samples were centrifuged at 2500× *g* for 10 min at 4 °C. After two washings, lipids were extracted with methanol/chloroform (1:2 *v/v*). The addition of NaCl 0.9% enabled the formation of hydrophilic (upper) and hydrophobic (lower) parts. The hydrophobic part was washed three times (1000× *g*, 10 min, 25 °C) before being added to scintillation vials. Lipids were concentrated via nitrogen flow. Radioactivity was measured using a scintillation counter (Tri-Carb 2800TR, PerkinElmer).

### 4.4. Lipidomics Analysis

The platelets were re-suspended in a modified Tyrode’s buffer with 0.35% free fatty acid BSA at a density of 10 × 10^5^/µL. Washed platelets (1.5 × 10^9^) were centrifuged at 11,700× *g* for 15 sec before being frozen. Lipid extraction and a lipidomics analysis were carried out, as previously described [7].

### 4.5. Extracellular Flux Analysis of Mitochondrial Respiration

PRP was centrifuged at 800× *g* for 10 min in the presence of PGE1 (100 ng/mL) and apyrase (1 U/mL). Platelet pellets were washed in modified Tyrode’s buffer without BSA containing apyrase (1 U/mL) and PGE1 (100 ng/mL). Washed platelets were re-suspended in modified DMEM’s medium (DMEM, 143mM NaCl, 0.5% phenol red, 5 mM D-glucose, 1mM sodium pyruvate, and 4 mM L-glutamine, pH 7.35) at a density of 2 x 10^6^/µL and incubated for 30 min at 37 °C with CP640.186. The 96-well plate was coated with 150 µg/mL “cell tak” solution (diluted in 0.1 M NaHCO_3_, pH 8.0) for 20 min at room temperature (RT) before being washed with distillated water. Overall, 15 × 10^6^ platelets were added to the 96-well plate, and their adhesion was induced by two centrifugations at RT (148× *g* for 1 min and 213× *g* for 1 min). The 96-well plate was incubated for 1 h 15 min in a CO_2_-free incubator before the assay. After basal oxygen consumption rate (OCR) measurement, 1 µM oligomycine, 0.45 µM FCCP, and 1µM rotenone/antimycin A mix were sequentially injected into the wells every 12 min. OCR was measured every 4 min using Seahorse extracellular flux XFp (Biosciences). Mitochondrial functional parameters were assessed via basal OCR, which was estimated by subtracting rotenone/antimycin A OCR from basal OCR, ATP-linked OCR, calculated by subtracting OCR after oligomycin from baseline, maximal OCR, reserve capacity, evaluated by subtracting basal OCR from OCR after FCCP, non-mitochondrial OCR, obtained after rotenone/antimycin A injection and proton leak, calculated by subtracting rotenone/antimycin A OCR from OCR after oligomycin.

### 4.6. Western Blotting

Washed human platelets (5 × 10^7^) were stimulated with 100 mU/mL thrombin for the indicated time points in the presence of 2 mM Ca^2+^ before being lysed in modified blue Laemmli’s buffer (50 mM Tris pH 6.8, 10% [*w/v*] glycerol, 2% [*w/v*] SDS, 0.5mM EDTA, 0.01% [*w/v*] bromophenol blue, and 25mM DTT). Whole platelet lysates were subjected to western blot. All antibodies were diluted in 5% BSA. The dilution for primary antibodies were (1/1000) except for anti-acetyl-α-tubulin (1/4000), anti-tubulin (1/10,000), anti-phospho-cofilin (1/10,000), anti-cofilin (1/20,000), anti-phospho-ERK (1/100,000), anti-ERK (1/2000), anti-gelsolin (1/50,000), and anti-GAPDH (1/100,000). Secondary HRP-conjugated anti-rabbit antibodies from Cell Signaling were diluted (1/5000), as were HRP-conjugated anti-rabbit and anti-mouse antibodies from Sigma (1/20,000). Band intensities were normalized to loading controls on the same gel. The respective total protein analyses were performed on different gels. Image J was applied to quantify all bands.

### 4.7. Platelet Aggregation and ATP Secretion

Aggregation and dense granules secretion were assessed using an aggregometer (Chrono-log 700, STAGO). Washed platelets (52.5 × 10^6^) were constantly stirred at 1200 rpm, 37 °C. Light transmission was recorded after CaCl_2_ (2 mM) addition and thrombin stimulation. To analyze ATP secretion, a luciferase/luciferin mix was added, which was followed by ATP standard.

### 4.8. Flow Cytometry Analysis

For CD62P exposure and αIIbβ3 activation, PGE1 (100 ng/mL) and apyrase (1 U/mL) were added to PRP before being centrifuged at 800× *g* for 10 min. Platelets were washed in a modified Tyrode’s buffer with 1.5% BSA. Washed platelets (6.25 × 10^6^) were incubated with anti-CD62P and PAC-1 antibodies (1/14.5) for 20 min at RT. Platelets were stimulated with thrombin 100 mU/mL at the indicated time points at 37 °C before being fixed with paraformaldehyde (PFA) 4%. For ROS production, washed platelets (6.75 × 10^6^) were re-suspended in a modified Tyrode’s buffer without BSA and incubated with 10µM H_2_DCFDA probe for 30 min at 37 °C. Platelets were washed two times with a modified Tyrode’s buffer without BSA containing apyrase (1 U/mL). Re-suspended platelets were stimulated with 100 mU/mL thrombin at different time points. The reaction was stopped by diluting platelets 14X with the modified Tyrode’s buffer without BSA. For F-actin content, washed platelets (6.25 × 10^6^) were re-suspended in the modified Tyrode’s buffer with 1.5% BSA. Re-suspended platelets were stimulated with 100 mU/mL thrombin at different time points at 37 °C before being fixed with 4% PFA. Platelets were permeabilized with 0.1% Triton and stained with FITC-conjugated phalloidin (10 µM) for 1 h at RT. Platelets were washed two times with phosphate buffer saline (PBS). All samples were analyzed using BD Canto II flow cytometer, and 10,000 events were recorded. As necessary, appropriate compensation controls were included.

### 4.9. Platelet Spreading and Immunofluorescence Staining

Washed platelets were re-suspended at a density of 3 × 10^4^/µL. Coverslips were coated with fibrinogen (100 µg/mL) overnight at 4 °C, blocked with PBS-BSA (5 mg/mL) for 1 h at RT and then washed with PBS. Platelets (6 × 10^6^) were either stimulated or not with 100 mU/mL thrombin in the presence of 2 mM CaCl_2_ and were immediately placed on coverslips for 15 min (actin) or 10 min (tubulin) at RT. Unstimulated platelets were fixed directly with 1% PFA and then added to coverslips. Non-adherent platelets were removed using a PBS wash. For actin visualization, adherent platelets were fixed with 1% PFA for 30 min at RT, permeabilized with 0.1% Triton, and stained with FITC-conjugated phalloidin (1:300) for 45 min at RT. To visualize the microtubule cytoskeleton, adherent platelets were blocked 30 min at RT before being incubated with anti-tubulin (1/800) and anti-acetyl-α-tubulin (1/100) antibodies for 1 h at RT. Coverslips were washed with PBS; anti-rabbit IgG coupled with alexa fluor 488 (1/500) and anti-mouse IgG coupled with alexa fluor 594 (1/1000) antibodies were added for 1 h at RT. Platelets were visualized using a Zeiss (x 100 oil immersion) fluorescence microscope (Carl Zeiss) equipped with ApoTome (AxioImager). The area (actin) covered by adherent platelets were quantified by means of Visiopharm software. Tubulin area was quantified using Image J.

## Figures and Tables

**Figure 1 ijms-22-13129-f001:**
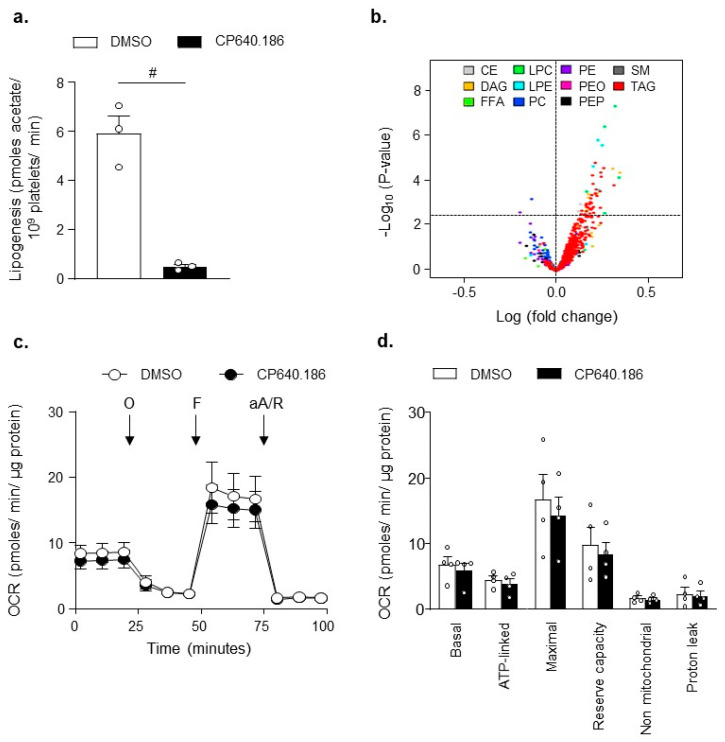
ACC inhibition induced by CP640.186 does not reduce overall lipid content and preserves mitochondrial function. (**a**) Washed human platelets were incubated with [1 − ^14^C]−acetic acid for 2 h in the presence of dimethyl sulfoxide (DMSO) or CP640.186 (60 μM). Radioactivity incorporation into lipid fractions was determined. Data were expressed as means ± SEM (*n* = 3); # *p*-value ≤ 0.0001 as compared to DMSO condition. Data underwent unpaired t-test. (**b**) Washed human platelets were preincubated with DMSO or CP640.186 (60 μM) before lipid extraction. Volcano plot representation of 672 detected lipid species. Lipids above the horizontal dotted line were up (right)− or down (left)−regulated with significant adjusted *p*-values ≤ 0.05. Log fold-changes and adjusted *p*-values were calculated from the multivariate regression model. CE = cholesterol ester; DAG = diacylglycerol; FFA = free fatty acid; LPC = lysophosphatidylcholine; LPE = lysophosphatidylethanolamine; PC = phosphatidylcholine; PE = phosphatidylethanolamine; PEO = plasmanyl phosphatidylethanolamine; PEP = plasmalogen phosphatidylethanolamine; SM = sphingomyelin; TAG = triacylglycerol. (**c**,**d**) Oxygen consumption rate (OCR) was measured in washed human platelets preincubated for 2 h with DMSO or CP640.186 before bioenergetic measurements. OCR was assessed under basal condition and after treatment with 1 µM oligomycin (O), 0.45 µM carbonyl cyanide p-trifluoromethoxyphenylhydrazone (F), and a mix of 1µM antimycin A (aA) and 1µM rotenone (R). (**c**) OCR profile is shown. (**d**) Mitochondrial function was evaluated by calculating basal, ATP-linked, maximum, non-mitochondrial OCR, reserve capacity, and proton leak. Results were expressed as means ± SEM (*n* = 4). The data were analyzed using unpaired t−test.

**Figure 2 ijms-22-13129-f002:**
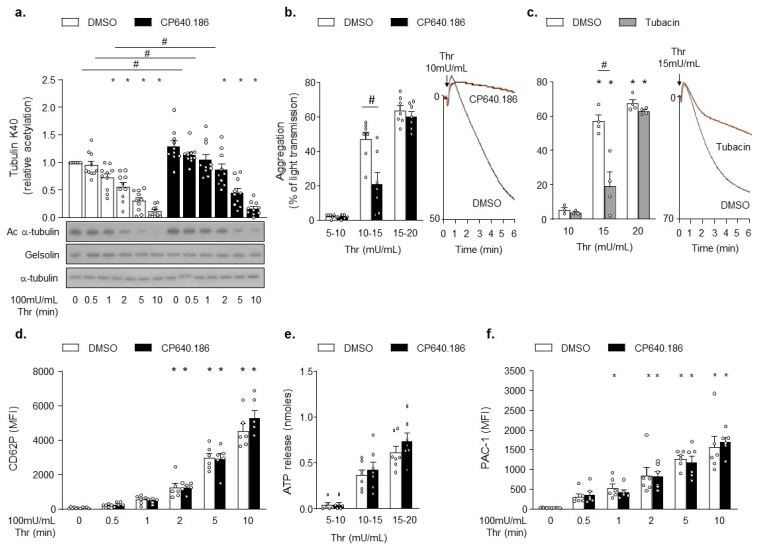
CP640.186-induced ACC inhibition increases tubulin acetylation levels and impairs platelet aggregation. (**a**,**b**,**d**,**f**) Washed human platelets were pre-incubated for 2 h with dimethyl sulfoxide (DMSO) or CP640.186 (60μM) or (**c**) for 1 h with DMSO or tubacin (10 µM), before being stimulated with thrombin (Thr). (**a**) Platelets were stimulated with 100mU/mL Thr at different time points. Whole platelet lysates were subjected to western blot analysis and probed with acetyl α−tubulin (Ac α-tubulin), α-tubulin, or gelsolin antibodies. Data were expressed as means ± SEM (*n* = 10); * *p*-value ≤ 0.05 in relation to unstimulated conditions. # *p*-value ≤ 0.05 in relation to DMSO conditions. Data were analyzed using 2−way ANOVA (analysis of variance). (**b**,**c**) Aggregation was analyzed by turbidimetry (Chrono-Log) in platelets stimulated with various Thr concentrations. Profile is shown on the right. Data were expressed as means ± SEM (*n* = 7). # *p*-value ≤ 0.0001 in relation to DMSO conditions. Data were analyzed using 2−way ANOVA. (**d**) P-selectin (CD62P) exposure was analyzed, by flow cytometry, in platelets stimulated with 100mU/mL Thr at various time points. Data were expressed as means ± SEM (*n* = 6). * *p*-value ≤ 0.05 in relation to unstimulated conditions. Data were analyzed using 2−way ANOVA. (**e**) Dense granule secretion was assessed following addition of luciferase-luciferin reagent in platelets stimulated with various Thr concentrations. Data were expressed as means ± SEM (*n* = 7). Data were analyzed using 2−way ANOVA. (**f**) Platelets were stimulated with 100mU/mL Thr at various time points, while αIIbβ3 activation (PAC-1) was detected by flow cytometry. Data were expressed as means ± SEM (*n* = 6). * *p*-value ≤ 0.05 in relation to unstimulated condition. Data were analyzed using 2−way ANOVA.

**Figure 3 ijms-22-13129-f003:**
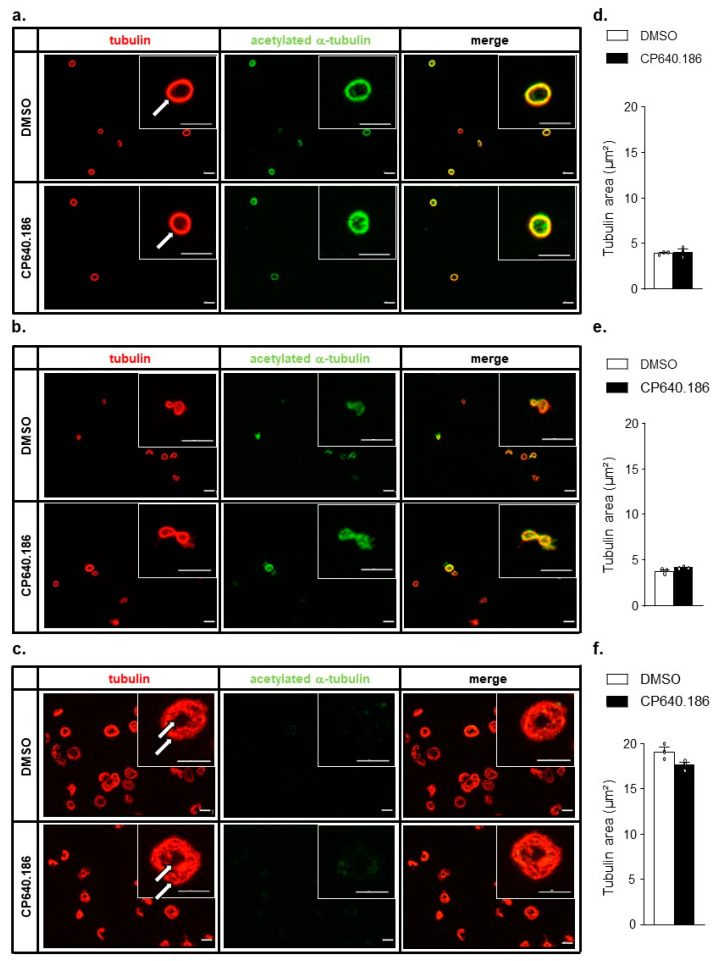
CP640.186-induced ACC inhibition does not affect microtubule dynamics. (**a**–**f**) Washed human platelets were pre−incubated for 2 h with dimethyl sulfoxide (DMSO) or CP640.186 (60μM) before being added to fibrinogen−coated coverslips for 10 min. (**a**) Unstimulated platelets were fixed directly using 1% PFA, before being added to coverslips. (**b**) Platelets were added to coverslips before being fixed with 1% PFA. (**c**) Platelets were stimulated with 100mU/mL thrombin and then added to coverslips before being fixed with 1% PFA. (**a**–**c**) Platelets were stained using tubulin (red) or acetyl α-tubulin (green) antibodies. Representative pictures are shown. Microtubule rings are indicated by white arrows. Scale bar: 5 µm. (**d**–**f**) Quantification of tubulin area. Data were expressed as means ± SEM (*n* = 3). Data were compared using unpaired *t*−test.

**Figure 4 ijms-22-13129-f004:**
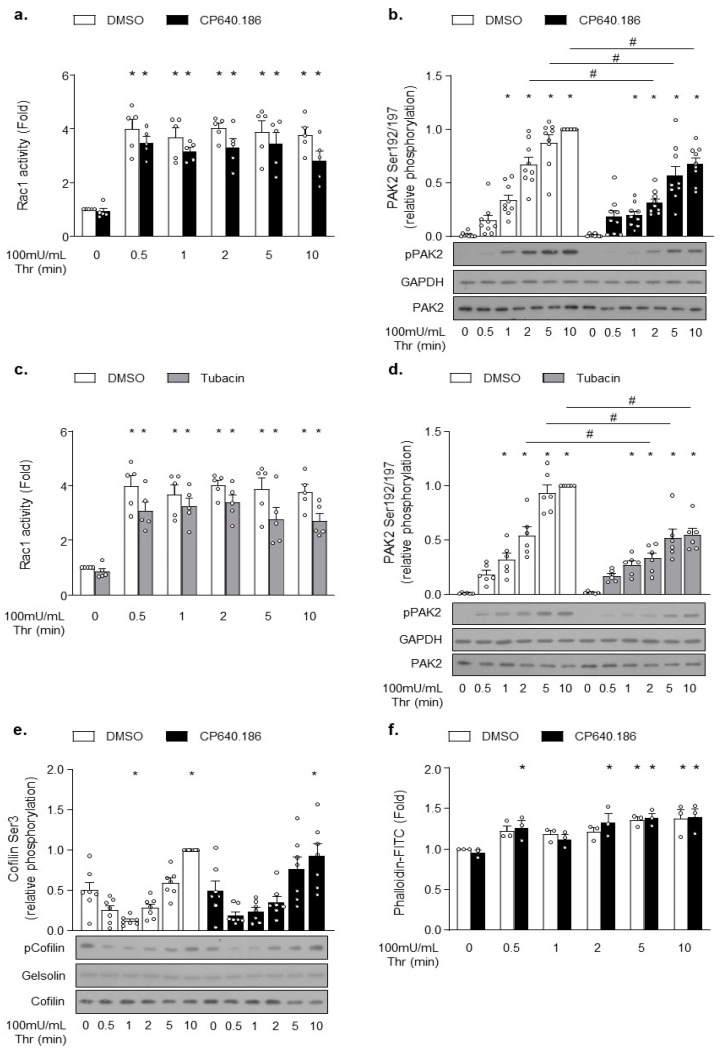
CP640.186-induced ACC inhibition reduces the Rac1−PAK2 pathway activation without impacting actin polymerization upon thrombin stimulation. (**a**,**b**,**e**,**f**) Washed human platelets were pre-incubated for 2 h with dimethyl sulfoxide (DMSO) or CP640.186 (60 μM) or (**c**,**d**) for 1 h with DMSO or tubacin (10 µM), before being stimulated with thrombin (100 mU/mL) at different time points. (**a**,**c**) Rac1 activity was assessed by means of G−LISA in whole platelet lysates. Data were expressed as means ± SEM (*n* = 5). * *p*-value ≤ 0.01 in relation to unstimulated conditions. Data were analyzed using 2-way ANOVA. (**b**,**d**,**e**) Whole platelet lysates were subjected to western blot and probed using phophoPAK, PAK2, GAPDH, phosphoCofilin, Cofilin, or gelsolin antibodies. Data were expressed as means ± SEM (at least *n* = 7). * *p*-value ≤ 0.05 in relation to unstimulated condition. # *p*-value ≤ 0.05 in relation to DMSO conditions. Data were analyzed using 2−way ANOVA (analysis of variance). (**f**) Platelets were stained with FITC−conjugated phalloidin (10 µM) for 1 h. F-actin content was analyzed by means of flow cytometry. Data were expressed as means ± SEM (*n* = 3). * *p*-value ≤ 0.05 in relation to unstimulated conditions. Data were analyzed using 2−way ANOVA.

**Figure 5 ijms-22-13129-f005:**
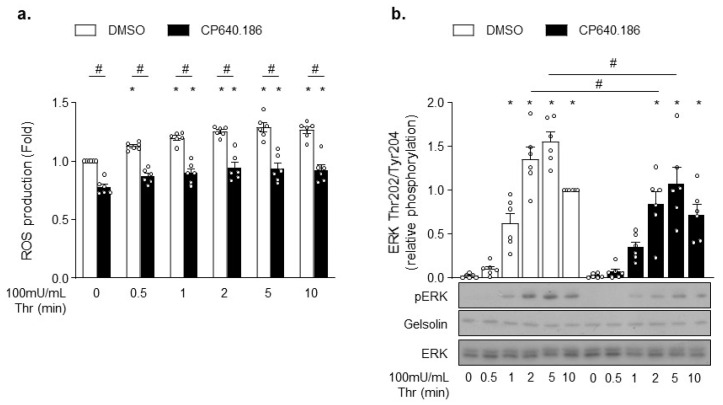
CP640.186 decreases thrombin-induced reactive oxygen species (ROS) production and ERK phosphorylation. (**a**,**b**) Washed human platelets were pre-incubated for 2 h with dimethyl sulfoxide (DMSO) or CP640.186 (60 μM) before being either stimulated or not with thrombin (100 mU/mL) at different time points. (**a**) Reactive oxygen species (ROS) were detected by means of flow cytometry using H_2_DCFDA (10 µM) probe. Data were expressed as means ± SEM (*n* = 6). * *p*-value ≤ 0.05 in relation to unstimulated conditions. # *p*-value ≤0.05 in relation to DMSO conditions. Data were compared using 2-way ANOVA (analysis of variance). (**b**) Whole platelet lysates were subjected to western blot and probed with phosphoERK, ERK, or gelsolin antibodies. Data were expressed as means ± SEM (*n* = 6). * *p*-value ≤ 0.05 in relation to unstimulated conditions. # *p*-value ≤ 0.05 in relation to DMSO conditions. Data were analyzed using 2−way ANOVA.

## Data Availability

Lipidomics data have been provided in Appendix A.

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
