# Peer review of "Acetyl-CoA Carboxylase Inhibitor CP640.186 Increases Tubulin Acetylation and Impairs Thrombin-Induced Platelet Aggregation"

_ijms, 2021, doi:10.3390/ijms222313129_

Round 1

Reviewer 1 Report

An excellent study, clearly and precisely written manuscript. The appropriate methodological approaches were undertaken to address the hypothesis put forward by the authors. Techniques were complementary to each other, and the results were well interrogated and interpreted. Data was well presented and explained. The referencing was supportive, relevant and topical. 

Only suggestion would be for future studies to maybe further clarify the effect on aggregation studies and the signalling modulation that may occur (given that there is little cytoskeletal remodeling-

  1. Does Acetyl-CoA carboxylase inhibitor CP640.186 effect avidity regulation of GPIIb/IIIa?
  2. What are the effects on Rap1b and CalDAG-GEF1 signalling axis.
  3. Are the G:F Actin ratio's affected
  4. A phospho-proteome analysis may throw further light to the precise molecular arbiters of the effects seen by CP640.186.

This would be food for thought for future studies. I applaud the authors on a very nice paper.

Reviewer 2 Report

The authors report that pharmacological inhibition (with CP640.186) of Acetyl CoA carboxylase (ACC) impairs thrombin-induced platelet aggregation through an increase in tubulin acetylation. They suggest that the underlying mechanisms involve a down-regulation of the Rac1-PAK2 signalling pathway and an associated decrease in ROS production and ERK-phosphorylation.

This is a well written and nicely presented paper with interesting, possibly important results on the regulation of protein/tubulin acetylation in human platelets. This may also be an import reminder to all platelet investigators that protein acetylation has to be considered as import regulatory PTM also in human platelets. There are, however, a few points the authors should consider in their current paper. They should be avoiding over-statements concerning causal relationships and also indicate the limitations of their present approach.

1) Almost all data rely on the use of the ACC inhibitor CP640.186. Although the authors published on ACC1 before, this is apparently the first publication reporting the use of CP640.186 with human platelets. Yet, the specific information on using this inhibitor is little. Why were the conditions reported chosen (2-hour preincubation, 60 µM concentration)? What are good positive/negative controls for their inhibitor experiments? What about non-target effects?

2) The authors argue that an inhibitor of tubulin deacetylase, tubacin, gave similar results supporting their conclusions. I suggest that the authors place the most important supporting data with tubacin in the main body of the paper, the remainder as before in the supplements. However, there are limitations here to compare causal effects of these two inhibitors.

3) The authors argue that CP640.186 induced tubulin acetylation plays an important role in their biochemical/ functional effects observed. How can the authors rule out that they see only correlation so far, and that many other acetylated proteins are involved? Have they compared  the acetylating capacity of acetylsalicylic acid (ASA) and 186 with human platelets?

4) The authors published previously that ACC is involve in the generation of  arachidonic acid-containing phosphatidylethanolamine plasmalogen lipids. Perhaps these or derivatives are involved in the functional effects observed in this paper.
